# Rigid Body Dynamics Simulation Based on GNNs with Constraints

## Abstract

In recent years, the utilization of Graph Neural Network (GNN)-based methods for simulating complex physical systems has opened new avenues for the fields of computational science and engineering. Despite their success, current GNN-based methods for rigid body dynamic simulation are constrained to relatively simple scenarios, hindering their practical use in industrial settings where complex mechanical structures and interconnected components prevail. These methods face challenges in handling intricate force relationships within rigid bodies, primarily due to the difficulty in obtaining force-related data for objects in industrial environments. To address this, we propose a novel constraint-guided method that incorporates force analysis into GNN-based simulations. The model incorporates computations related to both contact and non-contact forces into the prediction process. Additionally, it imposes physical constraints on the prediction process based on Kane's equations. We have rigorously demonstrated the model's rationality and effectiveness with thorough theoretical demonstration and empirical analysis. *Codes and anonymous links to the datasets are available in the supplementary materials.*

## 1 Introduction

Simulation of rigid body dynamics plays a pivotal role in numerous domains within numerical science and engineering Landau et al. (2008); Thijssen (2007); Szabó & Babuška (2021), including mechanical engineering Brach (2007), automotive design Tong (2000), biomechanics Silva et al. (1997), and virtual reality Sauer & Schömer (1998), enabling the accurate modeling and analysis of object motion, stress, and interactions for optimization and performance enhancement. Traditional methods for simulating rigid body systems often rely on intricate physical equations tailored to specific domains Huston & Passerello (1979); Mirtich (1996); Featherstone (2014). This necessitates a precise understanding of the physical properties of the analyzed objects and the physical environment in which they exist Thijssen (2007). Furthermore, such methods require a thorough grasp of relevant physical principles and relationships. With the rise of deep learning, data-driven methods have excelled in various fields by removing the need for detailed prior knowledge of system intricacies, significantly simplifying complex problems Dong et al. (2021). Recently, GNNs have transformed physics simulation, particularly in rigid body dynamics, by extending neural network applications to physical analysis Sanchez-Gonzalez et al. (2020); Pfaff et al. (2021). GNNs analyze objects modeled as graphs, demonstrating effective results in motion state analysis.

However, the current GNN-based methods concerning rigid body dynamic analysis are still limited to analyzing relatively simple and ideal scenarios, often focusing on the analysis of common geometric shapes Allen et al. (2023a); Bhattoo et al. (2022); Sanchez-Gonzalez et al. (2020); Han et al. (2022a); Bhattoo et al. (2022); Bishnoi et al. (2023); Allen et al. (2023b); Rubanova et al. (2021). This limitation hinders the practical application of such methods in industrial settings. In industrial environments, complex mechanical structures often involve numerous interconnected components, including intricate force relationships within rigid bodies and the resulting mutual influences between objects. Presently, GNN-based methods encounter significant challenges in addressing such problems, primarily because force-related data for objects is often challenging to observe and obtain Iglberger & Rüde (2011). As a result, these methods typically rely solely on positional and velocity information of objects for analysis. The difficulty in acquiring training data relevant to forces, coupled with the necessity for force relationship analysis, places these methods in a dilemma when applied to rigid

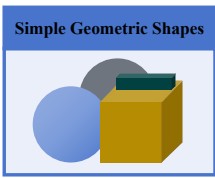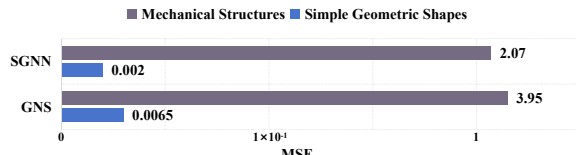

Figure 1: The comparison between the mechanical body scenarios we utilized and the general geometric body scenarios employed by other methods, results show that the performance of SGNN Han et al. (2022a) and GNS Sanchez-Gonzalez et al. (2020) exhibit significantly higher Mean Squared Error (MSE) in their predictive results for mechanical structures compared to geometric shapes.

body dynamics analysis with complex mechanical structures. The comparisons and experiments within Figure 1 validate such a conclusion, demonstrating a significant degradation in the performance of the relevant GNN-based methods for the analysis of complex mechanical structures.

To address this challenge, we endeavor to draw inspiration from traditional multibody dynamics analysis and incorporate relevant methods and constraints for calculating complex force relationships. It is noteworthy that while GNN-based dynamic analysis methods can assist in avoiding the need for acquiring detailed prior knowledge about the internal complexities of systems, the output results must adhere to the principles and rules of the physical world. Some researchers argue that a purely data-driven learning paradigm may struggle to achieve optimal results for physical problems Bhattoo et al. (2022); Han et al. (2022a); Bishnoi et al. (2023). Therefore, we aim to integrate the principles of traditional multibody dynamics analysis into GNN-based simulators, incorporating force analysis and prior constraints, to address the challenges associated with rigid body dynamics analysis in complex mechanical structures.

Building upon the aforementioned analysis, we propose a *Multibody Dynamics Guided GNN Simulator*, dubbed MDGS, to conduct a more thorough analysis of rigid body dynamics scenarios involving complex mechanical structures. Initially, we introduce a graph modeling pattern for efficient analysis of forces in physical systems, tailored for multibody dynamics. Subsequently, we introduce calculations related to forces, encompassing both contact and non-contact forces, into the methodology. These force calculations are performed using neural networks without the introduction of additional data beyond conventional GNN-based physical dynamics simulators. Then, the computed forces are integrated into the succeeding analyses to adapt to complex mechanical scenarios. Furthermore, we imposed constraints on the model output based on the Kane equations tailored for addressing multibody dynamics analysis problems. This ensures the accuracy and rationality of the model's computed results. The major contributions are as follows.

- We introduce a novel method, referred to as MDGS, to incorporate force analysis into GNN-based simulators for the state prediction of rigid body systems. This approach enhances the treatment of multibody dynamics scenarios, expanding the potential applications of such methods in industrial settings.

- We imposed physical constraints on our method based on Kane's equations, offering a more precise analysis of rigid body dynamics. These constraints are derived from rigorous theoretical analysis and proof.

- We provide the implementations of MDGS for multibody dynamics simulation tasks and create new datasets for validation, the results demonstrate the effectiveness of MDGS.

## 2 RELATED WORKS

**GNN-based Physical Dynamics Simulators.** GNN-based physical dynamics simulators Sanchez-Gonzalez et al. (2020) leverages GNNs Scarselli et al. (2009); Kipf & Welling (2017) to simulate predictive representations for graph-modeled physical systems. Such approaches find widespread applications in computing atomic forces Hu et al. (2021), simulating particle-modeled and mesh-modeled physical systems Allen et al. (2023b); Rubanova et al. (2021); Li et al. (2019); Sanchez-Gonzalez et al. (2020); Pfaff et al. (2021); Han et al. (2022b). In recent research, some methods Han et al. (2022a); Bhattoo et al. (2022); Bishnoi et al. (2023) have proposed incorporating prior knowledge from relevant domains into the GNN-based physical dynamics simulator to enhance its

performance. This includes integrating concepts such as Subequivariant and Lagrangian equations into the model framework, making the model more closely aligned with real-world physical systems. Additionally, some approaches Linkerhägner et al. (2023) optimize the model by collecting data in real-world environments as supplementary information. Our method introduces force analysis between physical entities in rigid body dynamics scenarios into the system and employs Kane's equations Kane & Levinson (1985) to impose constraints on the system output, ensuring a more realistic simulation performance.

**Dynamics Analysis.** The framework of traditional mechanical analysis relies on either force-based or energy-based approaches Wyk et al. (2022) for rigid body dynamics. Energy-based methods compute unknown energy using equilibrium equations, which is cumbersome for large structures Finzi et al. (2020). The force-based multibody dynamics model uses two recursive algorithms centered on the Newton–Euler formulation for forward kinematics and inverse dynamics Gonçalves et al. (2023). Some methods use ODEs to approximate optimal solutions for multibody conditions Schubert et al. (2023); Nada & Bayoumi (2023); Zhang et al. (2023). Traditional mechanical analysis requires precise physical parameters and domain knowledge. In contrast, GNN-based methods are data-driven and do not need these inputs. We combine elements of rigid body dynamics with GNNs for improved and simplified predictions.

## 3 METHODOLOGY

### 3.1 PRELIMINARY

**GNN-based simulators.** GNN-based simulators predict the dynamical states of physical systems. Consider a physical system comprising $M$ elements, collectively forming $N$ objects. At time $t$, GNN-based physical dynamics simulators model the aforementioned physical system using a graph $G^{(t)} = \{\mathcal{V}^{(t)}, \mathcal{E}^{(t)}\}$, where the node set $\mathcal{V}^{(t)}$ of graph $G^{(t)}$ represents different elements constituting objects. The feature $\boldsymbol{z}_i^{(t)}$ of node $i$ in the set $\mathcal{V}^{(t)}$ includes position information $\vec{\boldsymbol{x}}_i^{(t)}$, velocity information $\vec{\boldsymbol{v}}_i^{(t)}$, and the properties $\boldsymbol{k}_i$ of the object to which the element belongs, certain methods Allen et al. (2023a); Sanchez-Gonzalez et al. (2020) may retain only a subset of these features, such as velocity or position. $\boldsymbol{z}_i^{(t)}$ is defined as the concatenation of the vectors mentioned: $\boldsymbol{z}_i^{(t)} = [\vec{\boldsymbol{x}}_i^{(t)} || \vec{\boldsymbol{v}}_i^{(t)} || \boldsymbol{k}_i]$, where '$||$' denotes vector concatenation along the first dimension. $\boldsymbol{Z}^{(t)} = \{\boldsymbol{z}_i^{(t)}\}_{i=1}^M$ denotes the set of all $\boldsymbol{z}_i^{(t)}$. The edge set $\mathcal{E}^{(t)}$ of graph $G^{(t)}$ characterizes the connectivity relationships between elements. For instance, some approaches Han et al. (2022a); Bhattoo et al. (2022) judge that if the distance between element $i$ and $j$ is less than a certain threshold, the edge $(i, j)$ is connected. GNN-based simulator utilizes the GNN model $g(\cdot)$ to predict the values of $\boldsymbol{Z}^{(t+1)}$ based on $G^{(t)}$, such process can be formalized as $\boldsymbol{Z}^{(t+1)} = g(G^{(t)})$.

**Kane's Equation.** Kane's equation Kane & Levinson (1985) is a mathematical formulation used in the field of multibody dynamics to describe the motion of interconnected rigid bodies. For a nonholonomic system $\mathcal{S}$, the number of independently variable motions or deformations is referred to as degrees of freedom. For each degree of freedom, there exists a corresponding generalized velocity. These velocity vectors associated with generalized coordinates are typically denoted by $\dot{q}$, where $q$ represents the generalized coordinates. Specifically, with respect to the generalized coordinate $q_i$ and its corresponding generalized velocity $\dot{q}_i$, if $\dot{q}_i$ is independent of the other velocities in the system, meaning that it cannot be expressed or derived from other velocities in the system's description, then $\dot{q}_i$ is referred to as an independent velocity. Furthermore, the partial velocity $u'_{ij} = \frac{\partial \dot{q}_i}{\partial \dot{q}_j}$ denotes the partial derivative of the velocity with respect to coordinate $q_i$ when coordinate $q_j$ is varied, within the context of a multi-coordinate system. With the aforementioned concepts, we could define the general active force according to the $\gamma$-th independent velocity of $\mathcal{S}$ as follows:

$$K_\gamma = \sum_{i=1}^n \vec{F}_i \cdot \vec{u}'_{\gamma,i}, \tag{1}$$

where $\vec{F}_i$ denotes the force acting on the $i$-th mass point in the system $\mathcal{S}$, and $\vec{u}'_{\gamma,i}$ represents its partial velocity with respect to the $\gamma$-th independent velocity, $n$ is the number of mass points. Likewise, we

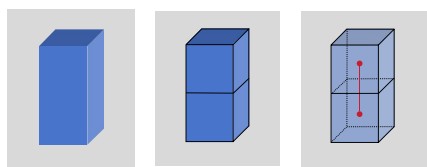

Figure 3: The framework of our proposed method. In the figure, the framework is segmented based on the stages of method execution, dividing it into three parts.

could define the generalized inertia forces according to the $\gamma$-th independent velocity as follows:

$$K_\gamma^* = \sum_{i=1}^{n} \left( -m_i \ddot{\vec{r}}_i \right) \cdot \vec{u}'_{\gamma,i}, \tag{2}$$

where $m_i$ represents mass, while $\vec{r}_i$ represents the vector radius. The Kane's equation can then be formulated as follows:

$$K_\gamma^* + K_\gamma = 0, \gamma = \{1, 2, ..., d\}, \tag{3}$$

$d$ is the number of independent velocities.

## 3.2 GRAPH CONSTRUCTION

Two commonly used object partitioning approaches in the field of physical simulation are point clouds Gross et al. (2002) and finite elements Pasciak (1995), both of which have been widely applied in GNN-based simulators. Point clouds essentially involve decomposing objects into indivisible elements using point cloud construction techniques. On the other hand, the finite element method divides objects into individual polyhedral elements, where the edges and vertices of these elements constitute the graph for GNN-based simulators. In our approach, we integrate the above two methods by employing the finite

(a)  Original (b) Partitioned (c) Graph Con-
Body      Body      struction

Figure 2: Illustration of graph construction.

element method to partition rigid bodies into different elements and designate the centroid of each element as a node in the graph. In the specific implementation, we assume uniform density within each element and utilize its geometric center as its centroid. We believe that this approach maximally preserves the kinematic properties of objects while aligning more closely with the commonly used analytical paradigm in multibody dynamics Kane & Levinson (1985). Figure 2 provides an illustrative example of our graph construction method. As illustrated in Figure 2(c), the centroids serve as nodes in the graph, and the centroids of adjacent elements are connected by edges. The specific methodology for edge construction will be elaborated in detail in Section 3.2.1.

### 3.2.1 CALCULATING FORCES

In the realm of multibody dynamics, the magnitude of forces acting on an object plays a crucial role in determining its state. However, due to the inherent difficulty in measuring and observing the force conditions of objects, existing GNN-based simulators do not currently incorporate force information into the prediction computation process. To address this issue, we employ neural networks to predict the forces acting on rigid bodies and integrate these predicted values into the model. Specifically, we categorize the forces acting on objects into two components: non-contact forces and contact forces.

For non-contact forces, such as gravity, we propose constructing an object-oriented graph $\widetilde{G}^{(t)} = \{\mathcal{V}^{(t)}, \widetilde{\mathcal{E}}^{(t)}\}$, where $\mathcal{V}^{(t)}$ represents the centroids corresponding to distinct elements, with attribute

features comprising the input positional and velocity information of these elements at time $t$. These features consist set $\boldsymbol{Z}^{(t)}$. Additionally, $\mathcal{V}^{(t)}$ includes property about the particular object each element is affiliated to, providing the information concerning the whole object. As for $\widetilde{\mathcal{E}}^{(t)}$, as illustrated in Figure 4(a), if two elements $i$ and $j$ belong to the same object and share faces between them, then an edge $(i, j)$ is established within $\widetilde{\mathcal{E}}^{(t)}$. Subsequently, we predict the set of non-contact forces $\widetilde{F}^{(t)}$. Formally, this prediction is expressed as:

$$\widetilde{F}^{(t)} = g^{\widetilde{F}}(\widetilde{G}^{(t)}), \tag{4}$$

where $g^{\widetilde{F}}(\cdot)$ denotes a GNN for non-contact force prediction, adopting the GCN described in Appendix B.2.

For contact forces, such as frictional and collision forces, we propose constructing a contact-oriented graph $\bar{G}^{(t)} = \{\mathcal{V}^{(t)}, \bar{\mathcal{E}}^{(t)}\}$. In $\bar{G}^{(t)}$, $\mathcal{V}^{(t)}$ remains identical to that of $\bar{G}^{(t)}$. The edge set $\bar{\mathcal{E}}^{(t)}$ comprises only those edges determined by the existence of contact between elements. As depicted in Figure 4(b), if the distance between the centroids of two elements, denoted as $i$ and $j$, is less than $r$ and they belong to different objects, we consider these elements to be in contact, and the edge $(i, j)$ is established, where $r$ is a threshold defined by a hyperparameter. In this manner, we derive a graph $\bar{G}^{(t)}$ specifically illustrating contact relationships.

Subsequently, based on the union of $\bar{G}^{(t)}$ and $\widetilde{G}^{(t)}$, we first obtain the features of each node to ensure that global information can be fully referenced when calculating contact-related forces. Formally, we have:

$$\bar{\boldsymbol{Z}}^{(t)} = g^{\bar{\boldsymbol{Z}}}(\bar{G}^{(t)} \cup \widetilde{G}^{(t)}), \tag{5}$$

Where $g^{\bar{\boldsymbol{Z}}}$ denotes a GNN for global information gathering. Afterwards, we employ $\bar{G}^{(t)}$ to predict the force set $\bar{F}^{(t)}$, where $\bar{F}^{(t)} \in \mathbb{R}^{N \times 3}$. The $i$-th vector $\bar{\boldsymbol{f}}_i^{(t)}$ of $\bar{F}^{(t)}$ denotes the contact forces acting upon the centroid of the $i$-th element. Formally, we have:

$$\bar{F}^{(t)} = g^{\bar{F}}(\bar{G}^{(t)}, \bar{\boldsymbol{Z}}^{(t)}), \tag{6}$$

where $g^{\bar{F}}(\cdot)$ denotes a GNN for contact force prediction. In selecting the specific GNN architecture, we opt for the conventional GCN Kipf & Welling (2017). Detailed information regarding network depth and hidden layer dimensions can be found in Appendix B.2. The vector $\bar{\boldsymbol{f}}_i^{(t)}$ will be set to the zero vector if $i$ is not an endpoint of any edge in $\bar{\mathcal{E}}^{(t)}$.

The final force set is obtained by combining the contact and non-contact force sets:

$$F^{(t)} = \{\boldsymbol{f}_i^{(t)} : \boldsymbol{f}_i^{(t)} = \bar{\boldsymbol{f}}_i^{(t)} + \widetilde{\boldsymbol{f}}_i^{(t)}\}, \quad (7)$$

where $\boldsymbol{f}_i^{(t)}$ and $\widetilde{\boldsymbol{f}}_i^{(t)}$ represent the $i$-th force vector of $F^{(t)}$ and $\widetilde{F}^{(t)}$ respectively.

### 3.3 MAKING PREDICTIONS

we predict $\boldsymbol{Z}^{(t+1)}$ based on $F^{(t)}$ and $G^{(t)}$. The edge set of $G^{(t)}$ equals $\widetilde{\mathcal{E}}^t$, as the contact forces are already in $F^{(t)}$, the contact edges $\bar{\mathcal{E}}^{(t)}$ are excluded. For the node set of $G^{(t)}$, we attach

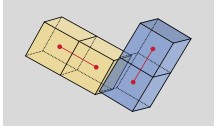 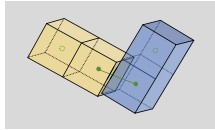

(a)  Object-oriented graph.  (b)  Contact-oriented graph.

Figure 4: Illustration of contact-oriented graph and object-oriented graph.

the calculated force to the node attribute $\boldsymbol{z}$. For the $i$-th node with attribute $\boldsymbol{z}_i$, the new attribute $\boldsymbol{z}_i'$ is calculated as follows:

$$\boldsymbol{z}_i' = [\boldsymbol{z}_i || \lambda F_i^{(t)}], \tag{8}$$

where $\lambda$ is a hyperparameter for controlling the magnitude of the influence of $F^{(t)}$. We then calculated the prediction as follows:

$$\boldsymbol{Z}^{(t+1)} = g(G^{(t)}), \tag{9}$$

where $g(\cdot)$ denotes the GNN for state prediction, adopting the same GNN architecture elaborated above. Motivated by Allen et al. (2023a), we introduced a rigid body structural constraint module in our method. This module ensures the invariance of rigid body structures in the predicted results. Specifically, we calculate the relative positions of various nodes on the object based on the initial values. Subsequently, we re-align our predicted results to the original relative positions, ensuring that the rigid body structure remains unchanged. A detailed description of this module can be found in **Appendix** B.1.

$\boldsymbol{Z}^{(t+1)}$ will then be utilized for calculating an MSE loss against the ground truth state $\boldsymbol{Z}^{*(t+1)}$ for the backpropagation of the whole framework:

$$\mathcal{L}_{mse} = \mathrm{MSE}(\boldsymbol{Z}^{(t+1)}, \boldsymbol{Z}^{*(t+1)}). \tag{10}$$

### 3.4 MODEL CONSTRAINTS BASED ON KANE'S EQUATION.

So far, we have incorporated force analysis to assist predictions within GNN-based simulators. However, due to the absence of ground-truth labels for the forces, these computations lack accuracy. As discussed in Section 3.1, Kane's equations provide physical constraints for rigid body dynamical systems. We aim to employ these constraints to refine our model, thereby enhancing its analytical fidelity to physical reality.

Theoretically, a model capable of accurately predicting rigid body systems adheres to the following principles:

**Theorem 3.1.** *Consider a rigid body dynamics system denoted as $\mathcal{S}$ with $d$ independent velocities. At the time $t$, the system's state is represented as $G^{(t)}$, constructed according to the detailed graph construction procedure outlined in Section 3.2 and 3.2.1. For any GNN denoted as $g(\cdot)$, which demonstrates accurate prediction capabilities for $F^{(t)}$ and $\boldsymbol{Z}^{(t+1)}$ based on $G^{(t)}$, the following equation must be satisfied:*

$$\sum_i F_{\gamma,i}^{(t)} u_\gamma - \sum_i M_i \frac{1}{\delta} \big( g(G^{(t)})_{[v_\gamma, i]} - v_\gamma^{(t)} \big) u_\gamma + \Psi^{(t)} = 0, \forall \gamma \in \{1, 2, ..., d\}, \tag{11}$$

*where $\gamma$ denotes the index of independent velocity, $F_{\gamma,i}^{(t)}$ is the predicted force value of element $i$ along $\gamma$-th independent velocity, $M_i$ is the mass of element $i$, $g(G^{(t)})_{[v_\gamma, i]}$ denote the predicted $v_\gamma^{(t+1)}$ with $g(\cdot)$, $\delta$ is the time step length, $\Psi^{(t)}$ is a disturbance term directly proportional to the sum of the radii from different mass points to the center of mass in each element. $u_\gamma$ represents the unit velocity value along the direction of the $\gamma$-th independent velocity.*

The proof is provided in **Appendix** A.1. Since Equation 11 in Theorem 3.1 holds for any model that accurately represents the target physical state, we can employ it to constrain the forces we have computed, thereby ensuring that our solution adheres more closely to the physical laws. Furthermore, Theorem 3.1 demonstrates that the constraints proposed in Equation 11 are valid along any independent velocity. Therefore, in practical implementation, we refine our model by adopting multiple constraint equations along different independent velocities. The number of equations the model needs to satisfy can thus be determined based on the distinct independent velocities associated with different systems.

To further constrain the perturbation terms in Theorem 3.1, we propose the following theorem:

**Theorem 3.2.** *Given the conditions within Theorem 3.1, then at time $t$, the following inequality holds:*

$$-\tau \sum_{i=1}^n (|\vec{\boldsymbol{R}}_i^{(t)}|)\omega_\gamma - \sum_{i=1}^n \frac{2}{5} M_i \tau^2 \dot{\omega}_i^{(t)} \leq \Psi^{(t)} \leq \tau \sum_{i=1}^n (|\vec{\boldsymbol{R}}_i^{(t)}|)\omega_\gamma + \sum_{i=1}^n \frac{2}{5} M_i \tau^2 \dot{\omega}_i^{(t)}, \tag{12}$$

*where $\vec{\boldsymbol{R}}_i^{(t)}$ denotes the interval force act on element $i$, $\tau$ is the max radius among all elements, $\omega_\gamma$ denotes unit angular velocity value along the direction of the $\gamma$-th independent velocity, $\dot{\omega}_i^{(t)}$ represents the max angular acceleration of element $i$.*

Based on Theorem 3.1 and Theorem 3.2, we proposed the following loss function for the upper bound of constrain:

$$\mathcal{L}_{sup} = \text{ReLU}\Big( \sum_i F_{\gamma,i}^{(t)} u_\gamma - \sum_i M_i \frac{1}{\delta} \big( g(G^{(t)})_{[v_\gamma,i]}$$
$$-v_\gamma^{(t)}\big) u_\gamma - \tau \sum_{i=1}^n (|\vec{R}_i^{(t)}|)\omega_\gamma - \sum_{i=1}^n \frac{2}{5} M_i \tau^2 \dot{\omega}_i^{(t)} \Big), \tag{13}$$

where $F_{\gamma,i}^{(t)}$ and $|\vec{R}_i^{(t)}|$ can be calculated with force prediction, while $\dot{\omega}_i^{(t)}$ is calculated based on variant within coordinates, $\text{ReLU}(\cdot)$ denote the ReLU function, $M_i$ is calculated based on the properties of the object, the details can be found in **Appendix B.1**. Similarly, we can calculate the loss function for the lower bound of constrain:

$$\mathcal{L}_{inf} = \text{ReLU}\Big( - \sum_i F_{\gamma,i}^{(t)} u_\gamma + \sum_i M_i \frac{1}{\delta} \big( g(G^{(t)})_{[v_\gamma,i]}$$
$$-v_\gamma^{(t)}\big) u_\gamma - \tau \sum_{i=1}^n (|\vec{R}_i^{(t)}|)\omega_\gamma - \sum_{i=1}^n \frac{2}{5} M_i \tau^2 \dot{\omega}_i^{(t)} \Big), \tag{14}$$

The overall loss for constrain is:

$$\mathcal{L}_{cst} = \mathcal{L}_{sup} + \mathcal{L}_{inf}. \tag{15}$$

$\mathcal{L}_{cst}$ will only be used to update $g^{\bar{F}}$ and $g^{\widetilde{F}}$. $\mathcal{L}_{cst}$ and $\mathcal{L}_{mse}$ are summed up for the final loss. The overall framework is illustrated within Figure 3.

## 4 EXPERIMENTS

### 4.1 ENVIRONMENTAL SETUP

**Dataset and Systems.** To validate the proposed MDGS, we conducted method verification on a variety of complex mechanical systems. Specifically, we utilized the Cubli robot Gajamohan et al. (2012) (a metallic mechanical cube capable of performing various maneuvers using three flywheels), complex rotating bodies (intricate mechanical structures featuring components such as bearings and hinges), vehicles (mechanical cars operating under different conditions and environments), UR5-Husky robot Wang et al. (2020) (vehicle-type robot equipped with mechanical arms), and 6-Dof Space robot Wang et al. (2022) (robot equipped with mechan-

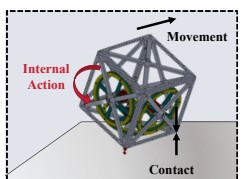 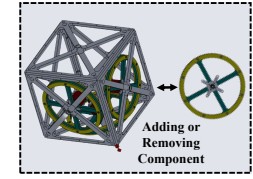

(a) Each dataset incorporates a variety of complex internal and external forces and motion states.

(b) Various adjustments are applied to the structures and actuators to create different datasets.

Figure 5: The dataset construction principles.

ical arms for executing specific tasks in space environments). We then modified the structures of these mechanical systems and introduced different environmental factors, resulting in the creation of 15 distinct datasets. Figure 5 provides a simple explanation of the datasets we constructed, with detailed descriptions available in **Appendix B.4**.

**Baselines.** We conduct a comparative analysis of our proposed MDGS against various baselines, including GNS Sanchez-Gonzalez et al. (2020), SGNN Han et al. (2022a), LGNS Bhattoo et al. (2022), and HGNS Bishnoi et al. (2023). Specifically, GNS employs a pure GNN for simulating physical systems, while SGNN introduces the concept of subequivariant in the physical system into the GNN-based physical dynamics simulator. Furthermore, LGNS and HGNS leverage Lagrangian equations and Hamiltonian mechanics, respectively, to guide and constrain the GNN model, enhancing its capability to simulate physical reality. Apart from GNS, the other three methods incorporate varying degrees of prior knowledge from physics to reinforce the models. Through comparisons with these baselines, we can effectively analyze the algorithmic performance of our approach in the dynamic analysis of complex mechanical bodies.

**Experimental Settings.** Our designed rigid body structural constraint module significantly enhances the accuracy of GNN-based simulators for rigid body problems. For a fair comparison, we integrated this module into other baseline models, improving their performance similarly. The effects of this integration are detailed in Section 4.3. We evaluated model performance using MSE between predicted and actual values, conducting each experiment 10 times to ensure statistical significance, as described in Han et al. (2022a). Details on specific hyperparameters and model backbones are provided in **Appendix** B.2, and computational complexity is discussed in **Appendix** B.3.

Table 1: Comparative experiment results on complex rotational body datasets and different robot datasets. **Bold** indicates the method with optimal performance. Underline denotes the method with second-best performance. Some standard deviations are marked as 0.00 due to being too small to represent effectively.

| | Senario | Cubli Robot | | | Complex Rotational Body | | | Space Robot | | |
|---|---|---|---|---|---|---|---|---|---|---|
| Time | Methods | Single Flywheel | Double Flywheels | Triple Flywheels | No Hinge | Side Hinges | Center Hinge | Single Arm | Double Arms | Under Strike |
| t=40 (MSE ×10⁻¹) | GNS | 0.89±0.17 | 0.32 ±0.26 | 0.75±0.05 | 7.77±1.79 | 0.80±0.09 | 0.65±0.19 | 6.5±0.00 | 11.2±0.52 | 31.24± 10.31 |
| | LGNS | 2.80±1.08 | 11.00±1.73 | 3.12±0.26 | 0.27±0.01 | 0.65±0.22 | 0.42±0.22 | 7.78±1.79 | 11.1±0.63 | 8.91±0.98 |
| | SGNN | 0.54±0.32 | 17.24±1.12 | 13.09±0.38 | 10.65±0.21 | 8.73±1.06 | 1.31±0.83 | 0.41±0.16 | 9.42±0.06 | 12.61±0.31 |
| | HGNS | 0.33±0.13 | 36.51±2.29 | 6.24±0.27 | 12.84±0.52 | 0.50±0.09 | 2.37±0.98 | 23.37±0.88 | 8.12±0.20 | 72.14±1.77 |
| | MDGS (Ours) | **0.17±0.08** | **0.24 ±0.06** | **0.31±0.03** | **0.15±0.04** | **0.11±0.08** | **0.18±0.04** | **0.39±0.00** | **3.22 ± 0.30** | **4.01 ± 0.36** |
| t=100 (MSE ×10⁻²) | GNS | 0.25±0.11 | 0.14±0.02 | 0.22±0.01 | 5.71±2.37 | 0.22±0.04 | 0.17±0.03 | 2.18 ±0.21 | 4.80 ±0.29 | 18.10 ±2.29 |
| | LGNS | 0.77±0.23 | 7.61±0.92 | 2.30±1.13 | 0.07±0.01 | 0.18±0.03 | 0.11±0.02 | 5.71±2.36 | 2.99±0.18 | 3.84 ±0.57 |
| | SGNN | 0.12±0.07 | 12.20±0.37 | 8.73±0.79 | 15.27±1.21 | 9.54±1.06 | 0.96±0.21 | 0.34±0.11 | 2.63±0.18 | 9.53 ±2.09 |
| | HGNS | 0.09±0.01 | 26.79±0.44 | 4.61±0.35 | 10.13±0.78 | 0.14±0.01 | 0.64±0.39 | 6.37±0.24 | 5.66±0.23 | 58.02 ±4.29 |
| | MDGS (Ours) | **0.05±0.01** | **0.06±0.01** | **0.09±0.01** | **0.04±0.01** | **0.03±0.00** | **0.05±0.00** | **0.22±0.00** | **2.40±0.33** | **2.51±0.93** |

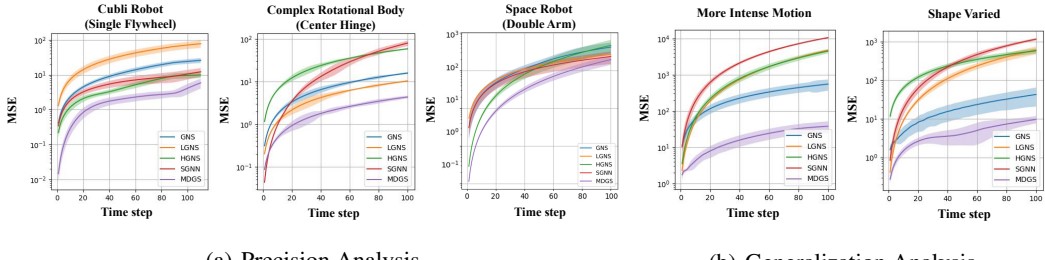

(a) Precision Analysis        (b) Generalization Analysis

Figure 6: Part of the experimental results on performance evaluation.

## 4.2 Performance Evaluation

**Precision.** Building upon the aforementioned experimental setup, we conducted performance testing experiments for both MDGS and other baseline algorithms across a total of 15 datasets. Table 1 and 2 presents the experimental results. It is evident from the results that our proposed MDGS consistently outperforms the majority of comparisons. Figure 6(a) demonstrated rollout-MSE curves on part of the datasets. We provide all the curves in **Appendix** B.5.

**Generalization.** We compared the generalization performance of various methods using the complete Cubli robot with three flywheels and introduced system attribute variations. The methods trained

Table 2: Comparative experiment results on vehicle datasets and UR5-Husky robot dataset. **Bold** indicates the method with optimal performance. Underline denotes the method with the second-best performance.

| | Senario | Vehicle | | | | UR5-Husky | |
|---|---|---|---|---|---|---|---|
| Time | Methods | Original | Fixed Obstacle | Damaged | Movable Obstacle | Original | Object Interaction |
| t=40 (MSE ×10⁻¹) | GNS | 1.52±0.58 | 7.77 ±1.79 | 5.81±0.33 | 60.15±0.69 | 28.81±0.33 | 163.61±16.22 |
| | LGNS | 0.39±0.07 | 4.39±0.07 | 0.41±0.16 | 88.33±0.89 | 10.60±1.30 | 219.07±35.00 |
| | SGNN | 0.45±0.00 | 1.85±0.00 | 0.39±0.05 | 23.11±0.31 | 82.01±5.58 | 154.44±11.98 |
| | HGNS | 0.21±0.03 | 9.99±0.03 | 2.79±0.08 | 108.25±2.82 | 6.97±0.13 | 172.02±9.95 |
| | MDGS (Ours) | **0.08±0.01** | **1.31±0.03** | **0.21±0.02** | **12.8±0.24** | **6.18±0.26** | **132.08±4.83** |
| t=100 (MSE ×10⁻²) | GNS | 0.37±0.15 | 5.71±2.36 | 4.06±0.08 | 304.99±0.38 | 19.47±6.67 | 44.44±4.43 |
| | LGNS | 0.07±0.02 | 3.54±0.12 | 0.34±0.10 | 114.52±0.14 | 7.35±1.97 | 59.40±9.72 |
| | SGNN | 0.25±0.00 | 1.29±0.01 | 0.11±0.01 | 24.14±0.01 | 53.81±1.35 | 41.82±3.07 |
| | HGNS | 0.08±0.01 | 6.73±0.01 | 0.77±0.23 | 159.15±3.89 | 5.10±0.18 | 46.57±2.69 |
| | MDGS (Ours) | **0.02±0.00** | **0.96±0.01** | **0.08±0.00** | **10.13±1.10** | **2.16± 0.03** | **35.68±1.28** |

Table 3: Generalization experiment results on Cubli robot dataset with triple flywheels. Δ indicates the disparity in model performance between the test set with changes and the original test set. **Bold** indicates the method with optimal performance. Underline denotes the method with second-best performance.

| Time | Methods | More Intense Motion | Δ | Less Intense Motion | Δ | Shape Varied | Δ | Density Varied | Δ |
|---|---|---|---|---|---|---|---|---|---|
| t=20 (MSE ×10⁰) | GNS | 115.61±18.08 | +113.53 | 5.41±0.27 | +3.33 | 5.01±0.62 | +1.06 | 8.17±3.57 | +4.22 |
| | LGNS | 188.71±28.27 | +157.41 | 40.79±13.93 | +9.49 | 21.77±0.83 | +13.25 | 30.72±9.43 | +22.20 |
| | SGNN | 614.13±31.04 | 566.36 | 97.02±49.25 | +49.26 | 57.53±1.02 | +21.26 | 61.92±48.41 | +25.65 |
| | HGNS | 216.76±47.45 | +119.74 | 197.58±61.12 | +100.56 | 40.53±1.02 | +23.86 | 120.01±34.54 | +103.34 |
| | MDGS (Ours) | **7.91± 2.85** | **+6.14** | **4.47±0.38** | **+2.71** | **2.13±0.26** | **+0.05** | **2.63±0.58** | **+0.55** |
| t=100 (MSE ×10²) | GNS | 6.08±0.05 | +5.97 | 0.36±0.14 | +0.22 | 0.31±0.03 | +0.09 | 0.48±0.26 | +0.26 |
| | LGNS | 58.33±4.91 | +50.72 | 11.60±0.15 | +3.97 | 6.41±1.32 | +4.11 | 7.25±0.16 | +4.95 |
| | SGNN | 124.04±11.04 | +111.83 | 26.80±2.47 | +1.45 | 13.11±2.37 | +4.38 | 13.67±0.02 | +4.94 |
| | HGNS | 54.40±7.15 | +28.61 | 50.86±8.32 | +24.07 | 9.76±0.98 | +5.15 | 6.37±0.24 | +1.76 |
| | MDGS (Ours) | **0.40±0.15** | **+0.33** | **0.25±0.02** | **+0.19** | **0.10±0.01** | **+0.01** | **0.11±0.02** | **+0.02** |

on the original dataset were tested for their accuracy on a modified system, with results shown in Table 3. Our method outperformed others, and notably, except for MDGS, methods incorporating prior physical system knowledge generally fared poorly in these experiments. Figure 6(b) demonstrated rollout-MSE curves on part of the datasets, all the curves are provided in **Appendix** B.5.

### 4.3    IN-DEPTH STUDY

**Visualization.**    To gain a comprehensive understanding of the performance of our proposed MDGS method, we visualized the prediction results in detail. As demonstrated in 7(a), it is evident that the MDGS method significantly enhances the accuracy of posture predictions for the Cubli robot when compared to the SGNN method. This improvement suggests that MDGS offers a more precise modeling approach for the internal flywheels and their intricate effects on rotational inertia, which are crucial for accurate dynamics simulation. Additionally, Figure 7(b) showcases MDGS's strong capabilities in trajectory prediction, highlighting its ability to anticipate movement patterns effectively. Meanwhile, Figure 7(c) further confirms the method's effectiveness when applied to complex robots operating within dynamic spatial environments, emphasizing its versatility and robustness. To enrich the understanding of our results, we also provide videos of the prediction outcomes in the supplementary materials, allowing viewers to see the model's performance in action and further illustrating the advantages of the MDGS method in real-time scenarios. These visualizations collectively reinforce our findings and demonstrate the ultimate goal of achieving enhanced predictive accuracy in robotic dynamics.

**Structural Constraint Module Analysis.**    To rigorously assess the effectiveness of the proposed structural constraint module, we conducted a series of validation experiments utilizing a Cubli robot dataset. This dataset provided a suitable platform to evaluate our model's performance under varying conditions. During these experiments, we compared the outcomes of our model with the structural constraint module against those obtained without it. As illustrated in Figure 7(e), the results demonstrate a marked improvement in model performance when the module is integrated, evidenced by significantly lower MSE values. This reduction in error highlights the module's efficacy in enhancing the accuracy of rigid body dynamics analysis. The detailed results, presented in digital form, can be found in the **Appendix** B.5, offering further insights into the quantitative improvements observed in our experiments. These findings underscore the importance of structural constraints in refining simulation fidelity and advancing the field of rigid body

**Ablation Study.**    We conducted a comprehensive series of ablation experiments aimed at isolating the effects of various constraints within Kane's equations. Specifically, we developed two baseline models, which we refer to as MDGS-NC and MDGS-SC, to evaluate the contributions of these constraints to the overall performance of our method. MDGS-NC is designed to remove all constraints from Kane's equations entirely, allowing us to assess the performance impact of a constraint-free environment. In contrast, MDGS-SC retains only an independent velocity constraint, providing a more limited framework for comparison. The experimental results, which are clearly illustrated in Figure 7(d), demonstrate a significant distinction between the performance of the three models. It is evident from our findings that the constraints inherent in Kane's equations play a critical role in enhancing the performance of our method, thereby validating our approach and suggesting that

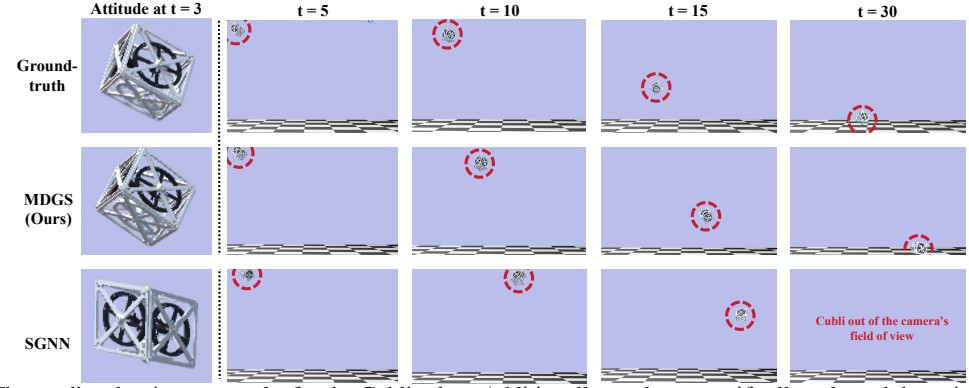

(a) The predicted trajectory results for the Cubli robot. Additionally, we have specifically enlarged the attitude of the Cubli robot at $t = 3$ on the left side for a detailed view.

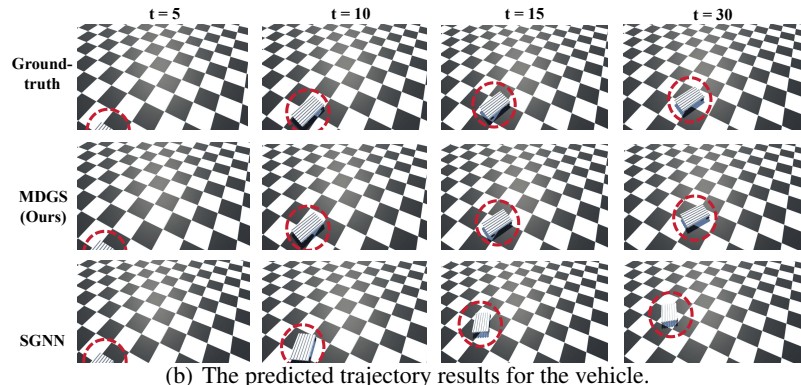

(b) The predicted trajectory results for the vehicle.

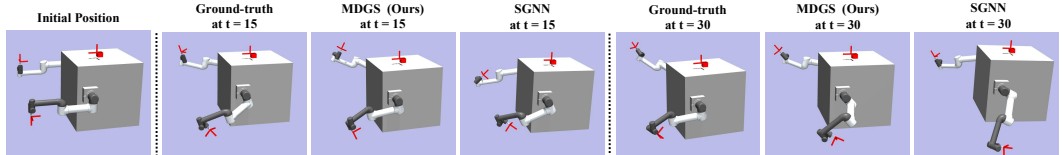

(c) The predicted results for the space robot.

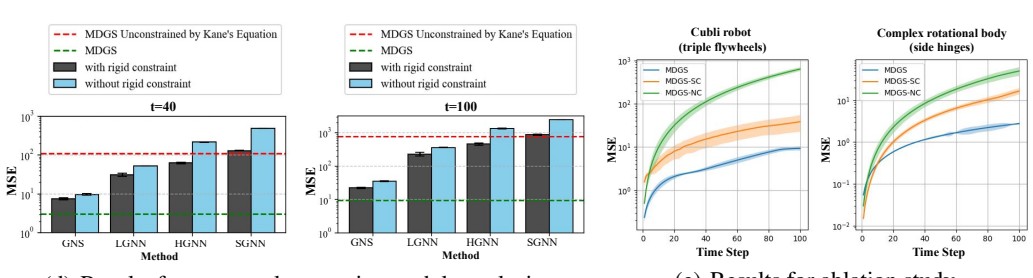

(d) Results for structural constraint module analysis.   (e) Results for ablation study.

Figure 7: In-depth study experimental results.

careful consideration of these constraints can lead to more accurate and robust simulations in complex mechanical systems.

## 5   CONCLUSION

In this paper, we propose a novel approach, MDGS, to incorporate force analysis into GNN-based rigid body system simulators, aiming to enhance the accuracy of such methods in complex mechanical scenarios. We substantiate the design of MDGS through theoretical foundations and proofs and validate its performance through a series of experiments.

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

## A  PROOFS

### A.1  THE PROOF OF THEOREM 3.1

We first present the following lemma to assist in establishing the rationality of the theorem.

**Lemma A.1.** *By partitioning each rigid body within system $\mathcal{S}$ into an element set $C = \{c_i\}_{i=0}^n$ following the procedure in Section 3.2, the motion states of all constructed elements within $C$, as well as the external forces $\{\vec{F}_i\}_{i=0}^n$ acting on these elements, conform to Kane's equation.*

*Proof.* According to the definition of generalized active force Kane & Levinson (1985), $\mathcal{S}$'s generalized active force $K_\gamma$ according to the $\gamma$-th independent velocity can be expressed as follows:

$$K_\gamma = \sum_{k=1}^m \vec{F}_k \cdot \vec{u}_{k,\gamma}, \tag{16}$$

where $m$ represents the number of point masses existing within $\mathcal{S}$, $\vec{F}_k$ denotes the external force vector acting on point mass $k$, $\vec{u}_{k,\gamma}$ denotes the partial speed corresponding to the $\gamma$-th independent speed. The $i$-th element can be regarded as a rigid that consists of multiple point masses. Therefore, we have:

$$
\begin{aligned}
K_\gamma &= \sum_{i=1}^n \sum_{l=1}^{\widetilde{m}_i} \vec{F}'_{i,l} \cdot (\vec{u}_{i,\gamma} + \vec{\omega}_{i,\gamma} \times \vec{r}_{i,l}) \\
&= \sum_{i=1}^n \sum_{l=1}^{\widetilde{m}_i} \vec{F}'_{i,l} \cdot \vec{u}_{i,\gamma} + \sum_{i=1}^n \sum_{l=1}^{\widetilde{m}_i} \vec{F}'_{i,l} \cdot (\vec{\omega}_{i,\gamma} \times \vec{r}_{i,l}) \\
&= \sum_{i=1}^n \Big( \sum_{l=1}^{\widetilde{m}_i} \vec{F}'_{i,l} \Big) \cdot \vec{u}_{i,\gamma} + \sum_{i=1}^n \Big( \sum_{l=1}^{\widetilde{m}_i} (\vec{r}_{i,l} \times \vec{F}'_{i,l}) \Big) \cdot \vec{\omega}_{i,\gamma} \\
&= \sum_{i=1}^n \vec{R}_i \cdot \vec{u}_{i,\gamma} + \sum_{i=1}^n \vec{M}_i \cdot \vec{\omega}_{i,\gamma}, \tag{17}
\end{aligned}
$$

where $\vec{F}'_{i,l}$ denote the external force acting on $l$-th point mass among the $\widetilde{m}_i$ point masses that consist element $c_i$, $\vec{u}_{i,\gamma}$ denote the partial velocity according to the $\gamma$-th independent velocity acting on the mass center of the element $c_i$, $\vec{\omega}_{i,\gamma}$ denote the partial angular velocity of element $c_i$ according to the $\gamma$-th independent velocity, $\vec{r}_{i,l}$ denotes the radius vector of the $l$-th point mass towards the mass center of the element $c_i$. $\vec{R}_i$ and $\vec{M}_i$ represent the forces and moments acting on individual elements. Such forces and moments can be calculated based on the properties of external forces $\{\vec{F}_i\}_{i=0}^n$. Consequently, we can derive that $K_\gamma$ can be expressed as aggregating the external forces and moments acting on these elements.

Furthermore, we can represent $\mathcal{S}'s$ generalized inertia force as follows:

$$K_\gamma^* = \sum_{k=1}^m (-m_k \vec{a}_k) \cdot \vec{u}_{k,\gamma}, \tag{18}$$

where $m_k$ denote the mass of the $k$-th point mass, $\boldsymbol{a}'_k$ represents its acceleration. Equation 18 can be reformulated as:

$$
\begin{aligned}
K_\gamma^* &= \sum_{i=1}^{n} \sum_{l=1}^{\widetilde{m}_i} (-m_{i,l} \vec{\boldsymbol{a}}_{i,l}) \cdot (\vec{\boldsymbol{u}}_{i,\gamma} + \vec{\boldsymbol{\omega}}_{i,\gamma} \times \vec{\boldsymbol{r}}_{i,l}) \\
&= \sum_{i=1}^{n} \sum_{l=1}^{\widetilde{m}_i} (-m_{i,l} \vec{\boldsymbol{a}}_{i,l}) \cdot \vec{\boldsymbol{u}}_{i,\gamma} + \sum_{i=1}^{n} \sum_{l=1}^{\widetilde{m}_i} (-\vec{\boldsymbol{r}}_{i,l} \times m_{i,l} \vec{\boldsymbol{a}}_{i,l}) \cdot \vec{\boldsymbol{\omega}}_{i,\gamma} \\
&= -\sum_{i=1}^{n} M_i \vec{\boldsymbol{a}}_i \cdot \vec{\boldsymbol{u}}_{i,\gamma} - \sum_{i=1}^{n} \sum_{l=1}^{\widetilde{m}_i} (\vec{\boldsymbol{r}}_{i,l} \times m_{i,l} \vec{\boldsymbol{a}}_{i,l}) \cdot \vec{\boldsymbol{\omega}}_{i,\gamma} \\
&= -\sum_{i=1}^{n} M_i \vec{\boldsymbol{a}}_i \cdot \vec{\boldsymbol{u}}_{i,\gamma} - \sum_{i=1}^{n} \vec{\boldsymbol{L}}_i^* \cdot \vec{\boldsymbol{\omega}}_{i,\gamma},
\end{aligned}
\tag{19}
$$

where $m_{i,l}$ denotes the mass of $l$-th point mass that consists element $c_i$, $\vec{\boldsymbol{a}}_{i,l}$ denotes the corresponding acceleration. $M_i$ and $\vec{\boldsymbol{a}}_i$ denotes the mass and the acceleration of $c_i$, $\vec{\boldsymbol{L}}_i^*$ can be determined based on the rotational state of $c_i$. Therefore, we can derive that $K_\gamma^*$ can be expressed as the aggregation of the motion states of elements within $C$. As demonstrated afore, $K_\gamma$ can be expressed as aggregating the external forces and moments acting on $C$. As both $K_\gamma^*$ and $K_\gamma$ within Kane's equation can be calculated with the given factors, we can conclude that the lemma holds. $\qquad\square$

Lemma A.1 demonstrates that the motion of element set $C$ and external forces $\{\vec{\boldsymbol{F}}_i\}_{i=0}^n$ conform to the Kane's equation. Next, we need to prove that equation 11 holds. According to the theorem, model $g(\cdot)$ is strictly accurate. Furthermore, from Equation 17, we have:

$$
\vec{\boldsymbol{R}}_i = \sum_{l=1}^{\widetilde{m}_i} \vec{\boldsymbol{F}}_{i,l},
\tag{20}
$$

which can be regarded as the resultant force acting on an element. Therefore, at time $t$, we have:

$$
\vec{\boldsymbol{R}}_i^{(t)} = [F_{1,i}^{(t)}, F_{2,i}^{(t)}, ..., F_{d,i}^{(t)}]^\mathsf{T},
\tag{21}
$$

where $F_{\gamma,i}^t$ denote the force value along coordinate $q_\gamma$ at time $t$. Based on the statements in the theorem, due to the independence of generalized velocities from each other, we could take the generalized velocity corresponding to each generalized coordinate as independent velocity, the partial velocity $\vec{\boldsymbol{u}}'_{i,\gamma}$ can be represented as follows:

$$
\vec{\boldsymbol{u}}'_{i,\gamma} = \frac{\partial \vec{\boldsymbol{r}}_i}{\partial q_\gamma} = \frac{\partial [q_{1,i}, ..., q_{d,i}]^\mathsf{T}}{\partial q_\gamma},
\tag{22}
$$

where $q_\gamma$ is the generalized coordinate corresponding to the $\gamma$-th independent velocity. $\vec{\boldsymbol{u}}'_{i,\gamma}$ is a one-hot vector in such circumstances. Therefore, we have:

$$
\vec{\boldsymbol{R}}_i^{(t)} \cdot \vec{\boldsymbol{u}}'^{(t)}_{i,\gamma} = [F_{1,i}^{(t)}, F_{2,i}^{(t)}, ..., F_{d,i}^{(t)}]^\mathsf{T} \cdot \frac{\partial [q_{1,i}, ..., q_{d,i}]^\mathsf{T}}{\partial q_\gamma} = F_{\gamma,i}^{(t)} u_\gamma,
\tag{23}
$$

where $u_\gamma$ represents the unit velocity value along the direction of the $\gamma$-th independent velocity. Therefore, at time $t$, we have:

$$
\begin{aligned}
K_\gamma^{(t)} &= \sum_{i=1}^{n} \vec{\boldsymbol{r}}_i \cdot \vec{\boldsymbol{u}}'^{(t)}_{i,\gamma} + \sum_{i=1}^{n} \vec{\boldsymbol{M}}_i^{(t)} \cdot \vec{\boldsymbol{\omega}}_{i,\gamma}^{(t)} \\
&= \sum_{i=1}^{n} F_{\gamma,i}^{(t)} + \sum_{i=1}^{n} \vec{\boldsymbol{M}}_i^{(t)} \cdot \vec{\boldsymbol{\omega}}_{i,\gamma}^{(t)} \\
&= \sum_{i=1}^{n} F_{\gamma,i}^{(t)} + \sum_{i=1}^{n} \sum_{l=1}^{\widetilde{m}_i} (\vec{\boldsymbol{r}}_{i,l} \times \vec{\boldsymbol{F}}_{i,l}^{(t)}) \cdot \vec{\boldsymbol{\omega}}_{i,\gamma}^{(t)}.
\end{aligned}
\tag{24}
$$

Similarly, we can also derive the following result:

$$K_\gamma^{*(t)} = -\sum_{i=1}^n M_i \vec{a}_i^{(t)} \cdot \vec{u}_{i,\gamma}'^{(t)} - \sum_{i=1}^n \vec{L}_i^{*(t)} \cdot \vec{\omega}_{i,\gamma}^{(t)}$$

$$= -\sum_{i=1}^n M_i a_{\gamma,i}^{(t)} u_\gamma - \sum_{i=1}^n \sum_{l=1}^{\widetilde{m}_i} (\vec{r}_{i,l} \times m_{i,l} \vec{a}_{i,l}^{(t)}) \cdot \vec{\omega}_{i,\gamma}, \tag{25}$$

where $a_{\gamma,i}^{(t)}$ denotes the accelerate speed value along coordinate $q_\gamma$ at time $t$. According to Kane's equation, we have:

$$K_\gamma^{(t)} + K_\gamma^{*(t)} = 0. \tag{26}$$

With Equation 24, 25, and 26, we have:

$$\sum_{i=1}^n F_{\gamma,i}^{(t)} u_\gamma + \sum_{i=1}^n \sum_{l=1}^{\widetilde{m}_i} (\vec{r}_{i,l} \times \vec{F}_{i,l}^{(t)}) \cdot \vec{\omega}_{i,\gamma}^{(t)} - \sum_{i=1}^n M_i a_{\gamma,i}^{(t)} u_\gamma - \sum_{i=1}^n \sum_{l=1}^{\widetilde{m}_i} (\vec{r}_{i,l} \times m_{i,l} \vec{a}_{i,l}^{(t)}) \cdot \vec{\omega}_{i,\gamma}^{(t)} = 0. \tag{27}$$

As the prediction model $g(\cdot)$ has assumed to be able to accurately predict the state of $\mathcal{S}$ in the Theorem 3.1, we have:

$$\sum_{i=1}^n F_{\gamma,i}^{(t)} u_\gamma - \sum_i M_i \frac{1}{\delta} \big( g(G^{(t)})_{[v_\gamma,i]} - v_\gamma^{(t)} \big) u_\gamma + \Psi^{(t)} = 0. \tag{28}$$

$\Psi^{(t)}$ is regarded as a disturbance term, formally:

$$\Psi^{(t)} = \sum_{i=1}^n \sum_{l=1}^{\widetilde{m}_i} (\vec{r}_{i,l} \times \vec{F}_{i,l}^{(t)}) \cdot \boldsymbol{\omega}_{i,\gamma}^{(t)} - \sum_{i=1}^n \sum_{l=1}^{\widetilde{m}_i} (\vec{r}_{i,l} \times m_{i,l} \boldsymbol{a}_{i,l}^{(t)}) \cdot \boldsymbol{\omega}_{i,\gamma}^{(t)} \tag{29}$$

From Equation 29, it can be easily conclude that:

$$\Psi^{(t)} \propto \sum_{i=1}^n \sum_{l=1}^{\widetilde{m}_i} |\vec{r}_{i,l}|. \tag{30}$$

So far, we have demonstrated Equation 11, along with the property of $\Psi^{(t)}$.

## A.2 THE PROOF OF THEOREM 3.2

Based on Equation 29, we have:

$$\Psi^{(t)} = \sum_{i=1}^n \sum_{l=1}^{\widetilde{m}_i} (\boldsymbol{r}_{i,l} \times \vec{F}_{i,l}^{(t)}) \cdot \boldsymbol{\omega}_{i,\gamma}^{(t)} - \sum_{i=1}^n \sum_{l=1}^{\widetilde{m}_i} (\boldsymbol{r}_{i,l} \times m_{i,l} \vec{a}_{i,l}^{(t)}) \cdot \vec{\omega}_{i,\gamma}^{(t)}. \tag{31}$$

Within Equation 31, the first term of $\Psi$ can be derived as follows:

$$\sum_{i=1}^n \sum_{l=1}^{\widetilde{m}_i} (\boldsymbol{r}_{i,l} \times \vec{F}_{i,l}^{(t)}) \cdot \vec{\omega}_{i,\gamma}^{(t)} \le \sum_{i=1}^n \sum_{l=1}^{\widetilde{m}_i} (|\boldsymbol{r}_{i,l}||\vec{F}_{i,l}^{(t)}| sin(\theta_{i,l}^{rF})) \omega_\gamma$$

$$\le \sum_{i=1}^n \sum_{l=1}^{\widetilde{m}_i} (|\boldsymbol{r}_{i,\arg\max_l(|\boldsymbol{r}_{i,l}|)}||\vec{F}_{i,l}^{(t)}| sin(\theta_{i,l}^{rF})) \omega_\gamma$$

$$= \sum_{i=1}^n |\boldsymbol{r}_{i,\arg\max_l(|\boldsymbol{r}_{i,l}|)}| \sum_{l=1}^{\widetilde{m}_i} (|\vec{F}_{i,l}^{(t)}| sin(\theta_{i,l}^{rF})) \omega_\gamma$$

$$\le \boldsymbol{r}_{\arg\max_i |\boldsymbol{r}_{i,\arg\max_l(|\boldsymbol{r}_{i,l}|)}|, \arg\max_l(|\boldsymbol{r}_{i,l}|)} \sum_{i=1}^n \sum_{l=1}^{\widetilde{m}_i} (|\vec{F}_{i,l}^{(t)}| sin(\theta_{i,l}^{rF})) \omega_\gamma, \tag{32}$$

where $r_{\arg\max_i |r_{i,\arg\max_l(|r_{i,l}|)}|,\arg\max_l(|r_{i,l}|)}$ is the max radius $\tau$ among all elements, $\theta_{i,l}^{rF}$ denotes the angle between $r_{i,l}$ and $\boldsymbol{F}_{i,l}^{(t)}$. The above expression can be further derived as follows:

$$\tau \sum_{i=1}^{n} \sum_{l=1}^{\widetilde{m}_i} (|\vec{\boldsymbol{F}}_{i,l}^{(t)}| sin(\theta_{i,l}^{rF}))\omega_\gamma = \tau \sum_{i=1}^{n} (|\vec{\boldsymbol{R}}_i^{(t)}| sin(\theta_{i,l}^{RF})) \leq \tau \sum_{i=1}^{n} (|\vec{\boldsymbol{R}}_i^{(t)}|)\omega_\gamma, \tag{33}$$

where $\theta_i^{RF}$ denote the angle between $\vec{\boldsymbol{R}}_i^{(t)}$ and $r_{i,l}$. Using the same method, we can also determine the lower bound of $\sum_{i=1}^{n} \sum_{l=1}^{\widetilde{m}_i} (r_{i,l} \times \vec{\boldsymbol{F}}_{i,l}^{(t)}) \cdot \vec{\boldsymbol{\omega}}_{i,\gamma}^{(t)}$, and thus we can obtain:

$$-\tau \sum_{i=1}^{n} (|\vec{\boldsymbol{R}}_i^{(t)}|)\omega_\gamma \leq \sum_{i=1}^{n} \sum_{l=1}^{\widetilde{m}_i} (r_{i,l} \times \vec{\boldsymbol{F}}_{i,l}^{(t)}) \cdot \vec{\boldsymbol{\omega}}_{i,\gamma}^{(t)} \leq \tau \sum_{i=1}^{n} (|\vec{\boldsymbol{R}}_i^{(t)}|)\omega_\gamma. \tag{34}$$

Meanwhile, the second term of Equation 31 can be derived as follows:

$$\sum_{i=1}^{n} \sum_{l=1}^{\widetilde{m}_i} (r_{i,l} \times m_{i,l}\vec{\boldsymbol{a}}_{i,l}^{(t)}) \cdot \vec{\boldsymbol{\omega}}_{i,\gamma}^{(t)} \leq \sum_{i=1}^{n} I_i \dot{\omega}_i^{(t)}, \tag{35}$$

where $I_i$ denotes the rotational inertia of element $c_i$. However, calculating the rotational inertia of irregular objects can be quite challenging, and this situation may arise with the elements we have divided. Therefore, we further derive Expressing 35 as follows:

$$\sum_{i=1}^{n} I_i \dot{\omega}_i^{(t)} \leq \sum_{i=1}^{n} \frac{2}{5} m_i \tau^2 \dot{\omega}_i^{(t)}, \tag{36}$$

where $\frac{2}{5} m_i \tau^2$ be the rotational inertia of a sphere that contains element $c_i$. Based on Equation 35 and 36, we have:

$$-\sum_{i=1}^{n} \frac{2}{5} m_i \tau^2 \dot{\omega}_i^{(t)} \leq \sum_{i=1}^{n} \sum_{l=1}^{\widetilde{m}_i} (r_{i,l} \times m_{i,l}\vec{\boldsymbol{a}}_{i,l}^{(t)}) \cdot \vec{\boldsymbol{\omega}}_{i,\gamma}^{(t)} \leq \sum_{i=1}^{n} \frac{2}{5} m_i \tau^2 \dot{\omega}_i^{(t)}, \tag{37}$$

Therefore, we have:

$$-\tau \sum_{i=1}^{n} (|\vec{\boldsymbol{R}}_i^{(t)}|)\omega_\gamma - \sum_{i=1}^{n} \frac{2}{5} m_i \tau^2 \dot{\omega}_i^{(t)} \leq \Psi \leq \tau \sum_{i=1}^{n} (|\vec{\boldsymbol{R}}_i^{(t)}|)\omega_\gamma + \sum_{i=1}^{n} \frac{2}{5} m_i \tau^2 \dot{\omega}_i^{(t)}, \tag{38}$$

The theorem is proved.

# B    EXPERIMENTAL DETAILS

## B.1    IMPLEMENTATIONS

### B.1.1    EDGE CONSTRUCTION FOR CONTACT-ORIENTED GRAPH.

In this section, we introduce the construction of edges in our CONTACT-ORIENTED GRAPH. Many physical scenarios are complex, involving multiple object interactions (including collision and friction) when the distance between them is minimal. The interactions between particles belonging to different objects and those within the same object are often distinct. Hence, we propose the concept of a contact graph. We continuously observe the distances between particles. When the distance between particles falls below a certain threshold, and they do not belong to the same object, we determine that contact has occurred. A new edge is established between these contacting particles to facilitate message passing, thus forming a new, contact-related graph. This newly constructed graph is utilized to predict and generate the contact forces experienced by the objects. Additionally, we leverage the pooling of particle-level information from the objects themselves to facilitate interactions between different objects.

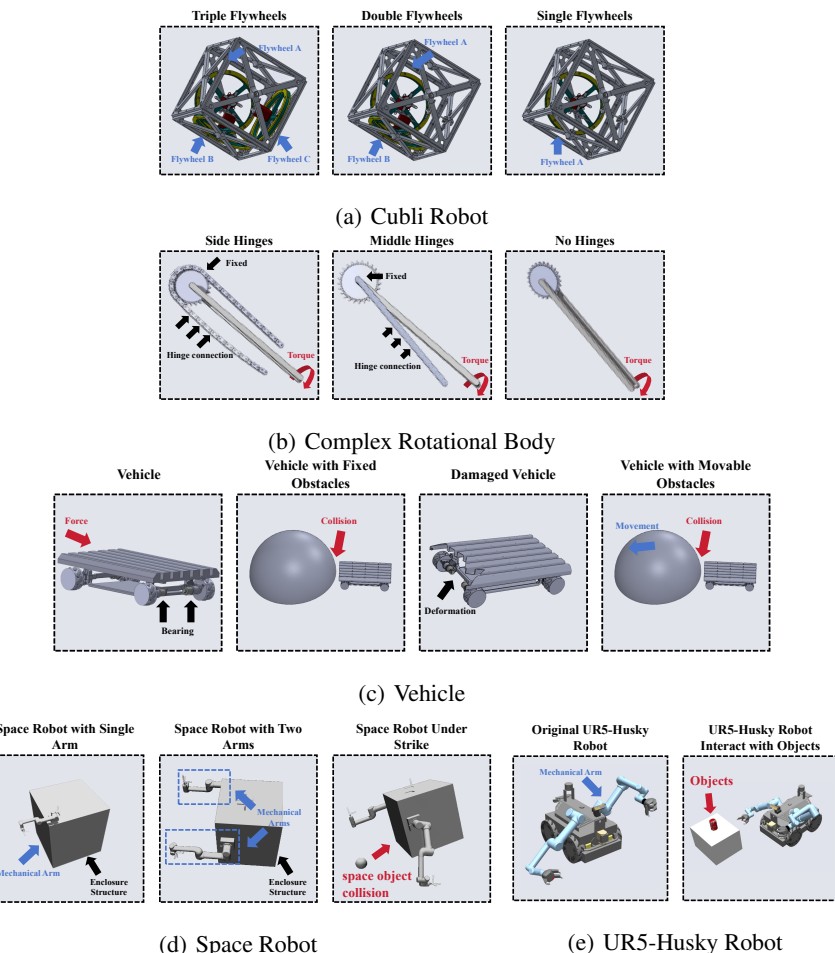

(a) Cubli Robot

(b) Complex Rotational Body

(c) Vehicle

(d) Space Robot      (e) UR5-Husky Robot

Figure 8: The physical systems used in all 15 datasets.

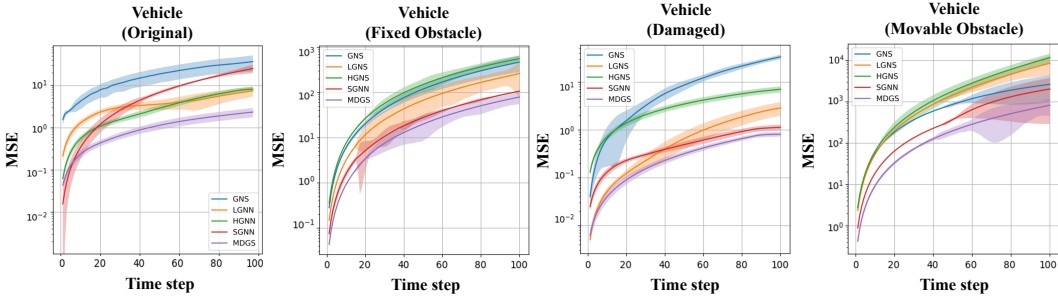

Figure 9: The rollout-MSE curves from the precision evaluation experiments conducted on vehicle datasets.

### B.1.2 THE SPECIFIC CALCULATION METHOD FOR EACH COMPONENT WITHIN $\mathcal{L}_{cst}$.

For the prediction of $F_{\gamma,i}^{(t)}$, it is essential to first clarify the number of independent velocities in our system. The rigid body system under study is a six-degree-of-freedom system, encompassing x, y, and z coordinates, as well as angular velocities along the x, y, and z axes. Nevertheless, given the complexity of acquiring rotational inertia related to angular velocity and the challenging nature of obtaining torque data, we limit our calculations to determining the magnitudes of force components associated with independent velocities along the x, y, and z directions. These force components can be directly obtained from the corresponding elements of the force vector $\boldsymbol{f}_i$ calculated in Section

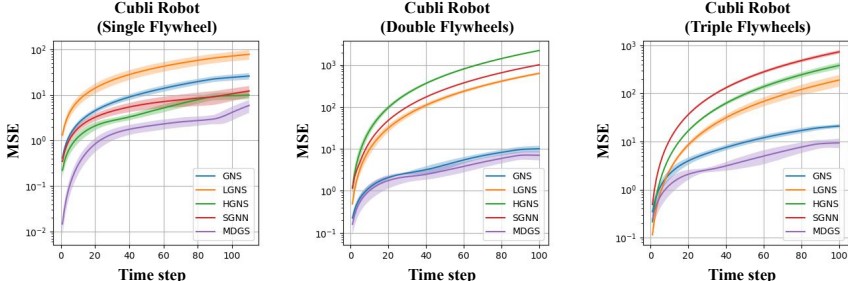

Figure 10: The rollout-MSE curves from the precision evaluation experiments conducted on Cubli robot datasets.

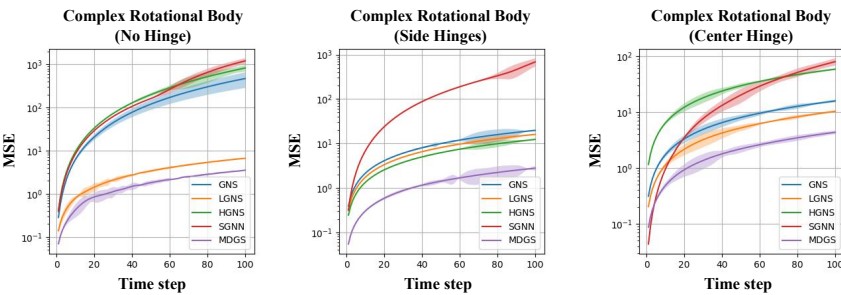

Figure 11: The rollout-MSE curves from the precision evaluation experiments conducted on complex rotational body datasets.

3.2.1. In this way, $|\vec{\boldsymbol{R}}_i^{(t)}|$ can also be directly obtained by calculating the magnitude of the force vector $\boldsymbol{f}_i$. $M_i$ can be obtained by calculating the volume of each element and multiplying it by the density.

We utilize the most commonly known laws of physics for solving angles issues. The angular velocity is derived by dividing the velocity difference of the current particle relative to the central particle by the distance between the current and the central particle. The formula is thus expressed as $\omega = \frac{v}{r}$, where $\vec{\boldsymbol{v}} = v_i(vec, present) - v_0(vec, center) = \sqrt{(v_{i,x} - v_{0,x})^2 + (v_{i,y} - v_{0,y})^2 + (v_{i,z} - v_{0,z})^2}$, $r = \sqrt{(x_i - x_0)^2 + (y_i - y_0)^2 + (z_i - z_0)^2}$. Similarly, for the velocity component of each axis, $\omega_{x,y,z} = \frac{v_{x,y,z}}{r_{x,y,z}}$, where $\vec{\boldsymbol{v_x}} = v_{i,x} - v_{0,x}, \vec{\boldsymbol{v_y}} = v_{i,y} - v_{0,y}, \vec{\boldsymbol{v_z}} = v_{i,z} - v_{0,z}; r_x = x_i - x_0, r_y = y_i - y_0, r_z = z_i - z_0$. We will select the angular velocity component along the largest axis to use as our angular velocity, $\omega_{max} = Max(\omega_x, \omega_y, \omega_z)$.

### B.1.3 STRUCTURAL CONSTRAINT MODULE

In this study, we introduce a novel algorithm aimed at restoring deformed objects to their original, regular rigid body shapes. This algorithm is based on the geometric transformation of three-dimensional point sets, achieving the restoration of the object's shape by precisely calculating the optimal rotation and translation matrices. Specifically, the algorithm selects elements from the current and reference states of the object for processing. It calculates the centroid of each set and centralizes the data, shifting the coordinates of both sets to be centered around the origin. This step eliminates the influence of positional deviations. Subsequently, it computes the covariance matrix of the two centralized data sets and employs Singular Value Decomposition (SVD) to extract the rotation matrix. To ensure the correct rotational direction of the object, we introduce a correction matrix to adjust the rotation matrix. This rotation matrix not only reflects the directional differences between the two point sets but also ensures that the transformation adheres to the right-hand rule. Finally, by calculating the translation vector, the adjusted reference point set is moved to a position as close as possible to the current point set. Our experimental results indicate that this method can effectively restore deformed objects back to their original shapes.

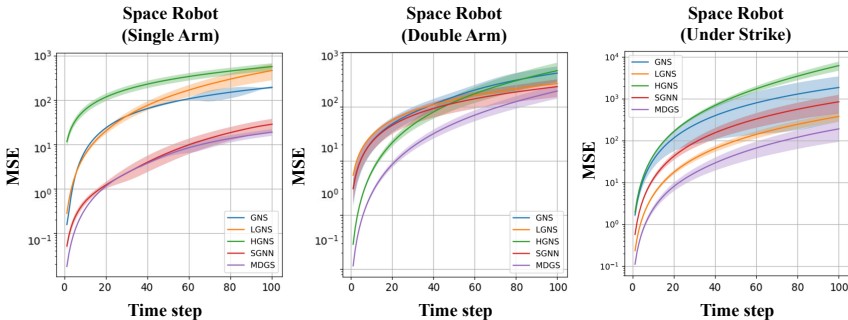

Figure 12: The rollout-MSE curves from the precision evaluation experiments conducted on space robot datasets.

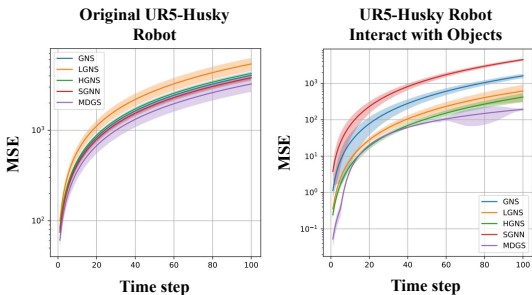

Figure 13: The rollout-MSE curves from the precision evaluation experiments conducted on UR5-HUSKY robot datasets.

## B.2 HYPERPARAMETERS AND SETTINGS

$g^{\widetilde{F}}(\cdot)$, $g^{\bar{F}}(\cdot)$, and $g(\cdot)$ utilize identical GCN architectures, employing ReLU as the activation function, and consist of three layers with a hidden dimension of 200. Due to its multi-stage hierarchical modeling, we use an Adam optimizer with an initial learning rate of 0.001 and beta values of (0.9, 0.999), along with a patience-based Plateau scheduler having 3 cycles and a decay factor of 0.8. Furthermore, we inject noise during training to achieve better long-term prediction at test time, with the noise ratio set to 4e-4 in Cubli, and 0.06 times the standard deviation in the other dataset. The cut-off radius $r$ for both datasets is set to 0.001. On these two datasets, we only use the state information of the last frame t as input to predict the information of frame t + 1. Experiments are conducted on a single NVIDIA RTX A6000 GPU.

## B.3 COMPUTATIONAL COMPLEXITY

Our algorithm exhibits a computational complexity akin to other GNN-based methods, specifically $\mathcal{O}(N \times d \times l)$. Here, $N$ and $d$ denote the number of nodes and degrees in the graph, respectively, while $l$ represents the number of layers. The complexity is primarily influenced by the node count. Our method constructs graphs with fewer nodes compared to other GNN-based approaches because it only utilizes centroids and does not require meshes or point clouds to represent regular shapes. In contrast, when examining traditional numerical simulation methods such as finite element analysis, their complexity is expressed as $\mathcal{O}(N' \times d' \times k')$, where $N'$ is the number of nodes, $d'$ the average degree, and $k'$ the number of functions computed. Like mesh-based methods, finite element analysis involves a larger $N'$, resulting in greater computational complexity.

## B.4 DATASET DETAILS

Utilizing Mujoco Todorov et al. (2012) for physics-based modeling, we systematically vary motion patterns and mechanical components to generate 15 datasets comprising 2200 trajectories, with

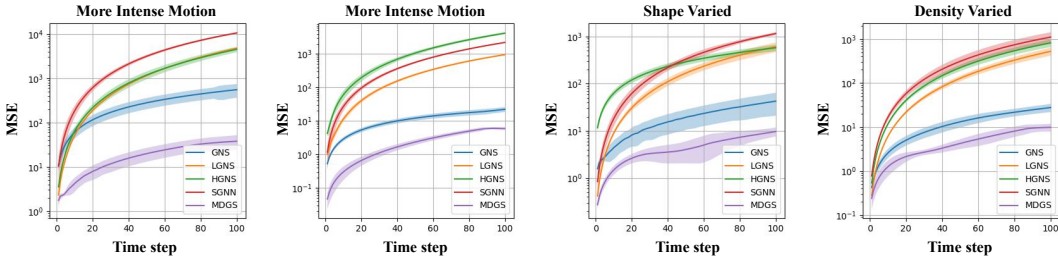

Figure 14: The rollout-MSE curves from the generalization evaluation experiments conducted on Cubli robot datasets.

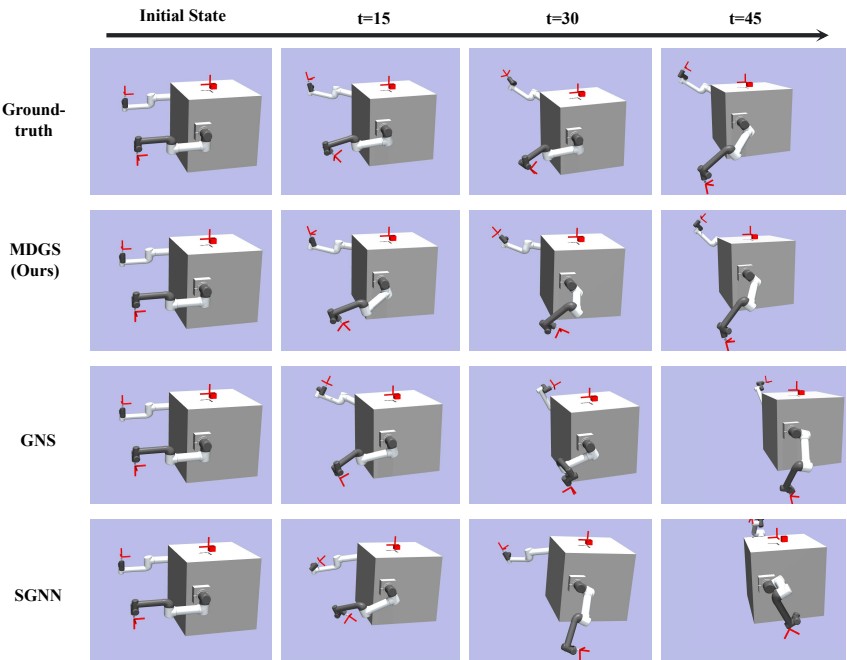

Figure 15: The complete visualization results of experiments upon space robot dataset.

each trajectory consisting of 110 time steps, enabling a thorough evaluation of the robustness of our approach. Figure 8 shows the detailed structures of all physical systems utilized in the experiments.

### B.5 FURTHER RESULTS

#### B.5.1 COMPARATIVE RESULTS

To more clearly display the results, we provide all the rollout-MSE curves here within Figure 13, 10, 11, 15, 13, and 14. From these results, it is evident that our method performs exceptionally well in most cases, significantly outperforming other comparative methods. This outcome robustly demonstrates the effectiveness of our proposed MDGS method and validates the necessity of incorporating physical knowledge into GNN-based simulators for accurate decision-making. By embedding physical knowledge into the GNN-based simulator, our approach can more accurately simulate and predict the behavior of complex systems, thereby enhancing overall performance.

We believe that the outstanding performance of MDGS across various scenarios is largely attributed to its successful integration of a force analysis framework into the model, along with the effective constraints based on principles of multibody dynamics. Specifically, the MDGS method considers the interactions among different components of the physical system during the modeling process, ensuring that the simulation results closely align with actual physical phenomena. Furthermore,

through the constraints imposed by multibody dynamics, our method is better equipped to handle multiple objects and their interactions within complex systems, thus maintaining high accuracy and robustness across diverse application scenarios.

### B.5.2 ABLATION STUDY RESULTS

Table 4: Ablation experiments compared with other baselines.

| Methods | Cubli Robot | | Complex Rotational Body | |
|---------|------|------|------|------|
| | t=40 | t=110 | t=40 | t=110 |
| GNS | 7.52±0.49 | 22.36±1.21 | 8.01±0.69 | 21.70±3.71 |
| LGNS | 31.2±2.66 | 230.09±31.94 | 6.52±0.87 | 17.59±2.52 |
| SGNN | 130.95±3.77 | 873.55±39.42 | 87.38±7.26 | 954.46±60.82 |
| HGNS | 62.44±2.68 | 461.5±35.41 | 4.98±0.19 | 13.53±1.07 |
| MDGS-NC | 110.84±12.47 | 763.36±1.46 | 8.40±0.33 | 60.15±7.05 |
| MDGS-SC | 15.24±5.88 | 43.47±18.94 | 3.32± 0.36 | 20.85±3.27 |
| MDGS | 3.11±0.32 | 9.59±1.27 | 1.15±0.31 | 3.04±0.64 |

Furthermore, Table 4 provides a more detailed comparison of ablation experiments. Results within the table present a numerical comparison of the methods used in our ablation experiments with other methods. These results further validate the rationale and necessity of our methodological framework.

### B.5.3 VISUALIZATION RESULTS

Figure 15 provides a clearer illustration of the action flow prediction results for the space robot. From the figure, we can observe that due to the complexity of the research subject, other baselines exhibit larger errors and tend to make significant misjudgments regarding the overall position. We believe this is because other methods lack stronger physical constraints, leading to greater inaccuracies when handling predictions for complex mechanical systems.

