# OpenReview forum: "Rigid Body Dynamics Simulation Based on GNNs with Constraints"
_ICLR.cc/2025/Conference — ICLR 2025 Conference Withdrawn Submission_

### Official Review · Reviewer_LQug · 2024-10-31

**Soundness:** 2
**Presentation:** 3
**Contribution:** 2
**Rating:** 3
**Confidence:** 3

**Summary:**

This paper predicts rigid body dynamics of physical systems using a Convolutional Graph Neural Network. It introduces two novel contributions: firstly, an architecture design which also predicts forces acting upon the objects, and secondly constrain the predictions based on Kane equations. Here, the predicted forces are further utilized. The authors compare their approach on in total 15 variations of five different datasets against other learned simulators. They further provide mathematical proofs for their work with respect to the constraints for the rigid body movement.

**Strengths:**

The paper presents a robust dataset benchmark with solid statistical validation for the proposed method. The release of both code and data provides a valuable resource for evaluating future advancements in rigid body simulation. Additionally, the focus on embedding physical constraints into network predictions addresses a relevant and impactful topic in the field.

**Weaknesses:**

- Figure 1 Analysis: The authors demonstrate that predicting mechanical structure generally yields higher errors than predicting geometric shape, concluding that the former is a harder task. However, I am unconvinced that comparing losses across different tasks substantiates this claim, given that variations in environmental settings can significantly impact loss scales. For example, have the authors ensured that these tasks are normalized in a strictly consistent way?

- Node Feature Representation: By using absolute node positions as node features in their graph input, the authors lose shift invariance, unlike prior work, such as the GNS baseline, which applies relative edge features for positional encoding. I wonder why the authors chose this approach and whether they have tested the more common relative positional encodings. A comparison here seems necessary.

- Baseline Implementation of GNS: The authors state they implemented the GNS baseline from Sanchez-Gonzalez et al. (2020), but they do not specify implementation details. From the code submission, it appears that the authors use a Graph Convolutional Network (GCN) with node features for positional encoding in this baseline. However, this is not the correct implementation, as the original work uses a Message Passing Neural Network (MPNN). Therefore, I do not believe the results accurately reflect a comparison against the state-of-the-art baseline. Although the proposed method shows strong results relative to other GCN approaches, this does not represent the top benchmark in neural simulation.

- Comparison to FIGNet: Besides the misimplementation of baselines, it would be critical to compare the method with FIGNet (Allen et al., 2023), currently one of the leading models for rigid body simulation.

- Graph Construction for Contacts: The authors construct two graphs—a contact-oriented graph and an object-oriented graph. I am uncertain whether the contact-oriented graph represents contacts meaningfully since each contact is treated as an independent graph component. For example, consider three objects, A, B, and C, where A contacts B and B contacts C. If A pushes B, this information cannot reach the contact between B and C (assuming B has multiple elements). Reducing the graph solely to contact edges also limits the receptive fields of the graph neural network. Have the authors considered an ablation study with a complete graph to model contact force and final prediction of $Z$?

- Force Accuracy Concerns: Given the lack of labeled force data, how can the authors ensure that the predicted forces $F_t$ align with the forces present in the multi-body experiment? The only learning signal directly affecting forces is from the constraints, but as far as I understand, these do not guarantee that predicted forces match real-world forces. Furthermore, Theorem 3.1 relies on accurate force predictions, meaning if the predicted forces don’t align, this undermines the theory presented in the paper. A comparison between predicted and ground-truth forces would clarify whether the model truly captures these forces. If I’ve misunderstood, I’d welcome clarification from the authors.

- Performance Evaluation: How does the runtime for a rollout using the proposed method compare to a ground-truth simulation based on physical principles? Rigid body simulations are highly optimized, so I wonder if neural simulation offers sufficient speed gains to justify the accuracy trade-off.

**Questions:**

The authors scale the input $F_t$ with a factor $\lambda$. Could they clarify the exact scaling applied here? Given that this input feeds into a neural network, minor variations in scaling should theoretically be mitigated by the network's initial weight matrix. What overall impact did this scaling factor have on the network's performance?

Minor typo:
- Line 228: [..] remains identical to that of $\bar G(t)$ [...]: I think it should be $\tilde{G}(t)$

---

### Official Review · Reviewer_PUqj · 2024-11-02

**Soundness:** 4
**Presentation:** 4
**Contribution:** 3
**Rating:** 8
**Confidence:** 3

**Summary:**

This paper presents MDGS, a novel Graph Neural Network (GNN)-based approach that integrates force analysis into rigid body simulations, enabling more accurate state prediction in complex multibody dynamics. By incorporating both contact and non-contact forces and enforcing physical constraints based on Kane’s equations, MDGS addresses limitations of previous GNN methods in handling intricate force relationships in industrial settings. The authors provide theoretical validation, empirical results, and new datasets, demonstrating MDGS’s effectiveness in simulating complex mechanical systems.

**Strengths:**

Originality: I am not aware of any prior work that specifically learns the dynamics of objects with complex mechanical structures, which is no easier than deformable objects. Figure 1 provides compelling evidence that previous approaches struggle to model such objects.

Quality: The theoretical proof in this paper is solid, although I have to admit that I don't fully understand the part talking about Kane's equation. The integration of force analysis and the construction of both object-centric and contact-centric graphs are well-conceived. The experimental results are thorough and convincing, both qualitatively and quantitatively.

Clarity: The paper is well-written and clear, with detailed explanations and intuitive figure illustrations that enhance understanding of the method.

Significance: The method outperforms previous approaches by a good margin in modeling objects with complex mechanical structures. Such objects have critical real-world applications; for instance, almost every robot falls into this category.

**Weaknesses:**

1. If I understand correctly, the MSE loss requires ground truth object states, which restricts the paper’s applicability to simulation scenarios. I’m curious if and how the authors plan to train the dynamics model using real-world data.

1. Additionally, I think one key advantage of neural dynamics models is their speed; however, the paper does not include an experiment comparing the speed of the proposed method with analytical methods. Could you provide runtime comparisons between MDGS and traditional analytical methods (e.g., finite element analysis) for simulating complex mechanical systems? This would give readers a clearer picture of the computational trade-offs.

1. A more comprehensive discussion of the method’s limitations from the authors would be appreciated.

**Questions:**

1. Are there any limitations to the proposed graph construction method, which includes both an object-centric graph and a contact-centric graph, or do you consider this a good approach for modeling multi-rigid-body dynamics in general?

1. Is the proposed method restricted to certain types of mechanical systems? For instance, what challenges might arise when learning the dynamics model of a humanoid robot?

---

### Official Review · Reviewer_8WU4 · 2024-11-04

**Soundness:** 3
**Presentation:** 3
**Contribution:** 3
**Rating:** 5
**Confidence:** 4

**Summary:**

The paper addresses the limitations of existing GNN methods for simulating complex rigid body dynamics in industrial applications. The authors propose a novel constraint-guided approach that incorporates both contact and non-contact forces, enhancing prediction accuracy. By applying physical constraints from Kane’s equations, the model improves overall MSE across various datasets.

**Strengths:**

The paper provide a possible solution of properly incorporating physical constraints in simulation rigid body dynamics, which are crucial for enhancing the accuracy and realism of these simulations.

**Weaknesses:**

The paper lacks a more detailed description of the datasets and experimental settings, as well as a thorough evaluation of the results. This omission makes it challenging to fully understand the context and implications of the findings. Detailed questions and comments are listed below.

**Questions:**

1.  Is there an ablation study on other GNN architectures besides `GCN`? If the authors utilize `GNS`  as the foundational structure instead, the performance depicted in Figure 7(d) may not be accurate (i.e. MDGS will have the same performance as `GNS+rigid constraint`). Also, as adding structure constraint module could improve the performance of all baseline methods, the main contribution of this paper seems to be the introduction of a model-agnostic structural constraint module.
2. The paper lacks detailed descriptions of each dataset. For example, how control variables vary between the training and testing datasets? How many number of nodes or objects are there in each dataset?
3. Moreover, there is insufficient information about the partitioning method and the resulting partitions for each dataset. An ablation study on the partitioning method would be beneficial. Visualizations of the partitions would also be helpful. Also, the size of the partitioned objects is significant, as too few may not provide enough information, while too many could lead to redundancy. Any ablation studies on this aspect?
4. Although there are no ground-truth labels for the forces, it is still possible to analyze the relative learned values among these forces across the datasets. For instance, examining the relative magnitudes of learned forces at different locations and at different time steps could yield valuable insights.
5. Solely comparing the MSE does not adequately verify the validity of the added physical constraints. The authors should utilize some metric to evaluate whether the constraints are more effectively satisfied by MDGS compared to other methods.
6. In tables such as Table1, does $t=40$ represent the MSE at that specific time step, or it is the average MSE for all time steps $t \leq 40$?

---

### Official Review · Reviewer_xisR · 2024-11-04

**Soundness:** 2
**Presentation:** 1
**Contribution:** 1
**Rating:** 3
**Confidence:** 5

**Summary:**

The paper introduces a GNN architecture for learning rigid body dynamics. The paper attempts to connect Kane's equations with the architectural design choices of the GNN. The approach is validated on simulated datasets.

**Strengths:**

- **Modeling closed-loop systems is valuable**. I appreciate the authors' efforts in modeling the Cubli robot. These systems, seemingly closed-loop, add value to the evaluations of the approach.
- **Generalization experiments add value**. I appreciate the authors' efforts in evaluating their approach with varying shapes and different intensity levels of motions and density parameters.

**Weaknesses:**

**A. Poor presentation**.

Frankly, the paper's presentation is below the standard for ICLR. In terms of writing clarity, I am especially concerned about the authors' introduction of Kane's Equation, starting at line 146. The section on Kane's equation, supposedly the most relevant component of this work, lacks motivation and connection to GNN designs made later. As a reader, this section distracts me from the bulk of the work and feels taken from a textbook and inserted into the paper without context.

Next, the teaser figure is highly confusing. As the first figure that the readers see, I would recommend the authors consider what separates their work from GNS or SGNN cited in the paper (perhaps Kane's equations aspect) and communicate that clearly to the readers.

It seems that several crucial implicit assumptions are not clarified in the paper. For example, the coordinate system used in the paper seems to lack transparency and confuses me. How are the graphs of Euclidean coordinates (particles in this case) constructed under the view of the generalized coordinate system that has been presented throughout the whole paper? If the technique described in the paper requires expert knowledge of the minimal coordinate systems (e.g., writing articulated systems as in joint coordinates) to construct the graph, then the authors must clarify that more clearly. If that is indeed the case, it then confuses me why don't we just use a first-principled rigid-body physics simulator, such as Drake [1], in this case. What's the value of learning a simulator when you assume knowledge of the minimal coordinate systems?

[1] Russ Tedrake and the Drake Development Team, Drake: Model-based design and verification for robotics, 2019.


**B. Questionable rigor of physics**.

Even though the paper advertises a "rigorous" treatment of physics, I find the rigor of the work not so compelling. This is related to my comments on presentation clarity. If Kane's equations are derived from the variational principle, the authors must state that. Have the authors visualized the F_tilde and F_bar against the model proposed in Kane? Currently, the section on Kane's equations seems to me like an attempt to make the architecture proposed in this work look physics-based when it is not.

The authors also portray the boundary conditions under the umbrella of "non-contact" forces. How does the proposed system represent boundary conditions, such as contacts with the floor, and more broadly?

What quantities are conserved, if at all? Can the authors visualize if the proposed learning system adds or removes energy?

**C. Missing critical experiments**.

As I mentioned above, if the motivation is to connect with Kane's equations, then the authors need to surgically visualize the components of their model and convince the audience that the transformations learned by the model match the definitions in Kane's equations, by visualizing predicted forces on the particle level. I can't seem to find such visualizations in the main manuscript.

**Questions:**

Please see my comments in the weakness section. Thanks.

---

### Note · Authors · 2024-11-16

I have read and agree with the venue's withdrawal policy on behalf of myself and my co-authors.